



# Rescue and homogenisation of 140 years of glacier mass balance data in Switzerland

Lea Geibel[1,2], Matthias Huss[1,2,3], Claudia Kurzböck[1,2], Elias Hodel[1,2], Andreas Bauder[1,2], and Daniel Farinotti[1,2]

[1]Laboratory of Hydraulics, Hydrology and Glaciology, ETH Zürich
[2]Swiss Federal Institute for Forest, Snow and Landscape Research (WSL), Birmensdorf, Switzerland
[3]Department of Geosciences, University of Fribourg, Fribourg, Switzerland)

**Correspondence:** Matthias Huss (huss@vaw.baug.ethz.ch)

**Abstract.** Glacier monitoring in Switzerland has resulted in some of the longest and most complete data series globally. Mass balance observations at individual locations, starting in the 19th century, are the backbone of the monitoring as they represent the raw and original glaciological data demonstrating the response of snow accumulation and snow/ice melt to changes in climate forcing. So far, however, the variety of sources of historic measurements has not been systematically processed and

documented. Here, we present a new complete and extensive point glacier mass balance dataset for the Swiss Alps that provides attributes for data quality and corresponding uncertainties. Original sources were digitized or re-assessed to validate or to correct existing entries and to identify metadata. The sources of data are highly diverse and stem from almost 140 years of records, originating from handwritten field notes, unpublished project documents, various digital sources, published reports, as well as meta-knowledge of the observers. The project resulted in data series with metadata for 63 individual Swiss glaciers,

including more than 60'000 point observations of mass balance. Data were systematically analyzed and homogenized, e.g. by supplementing partly missing information based on correlations inferred from direct measurements. A system to estimate uncertainty in all individual observations was developed indicating that annual point balance is measured with a typical error of 0.07 m water equivalent (w.e.), while the average error in winter snow measurements is 0.20 m w.e. Our dataset permits further investigating the climate change impacts on Swiss glaciers. Results show an absence of long-term trends in snow accumulation

over glaciers, while melt rates have substantially increased over the last three decades.

## 1   Introduction

Mass loss of mountain glaciers in the Alps and worldwide in response to a changing climate is rapid and accelerating (Zemp et al., 2019; Hugonnet et al., 2021). Understanding its dynamics and drivers is a necessary precondition to assess the glaciers' role as present and future water resources (e.g. Van Tiel et al., 2018; Immerzeel et al., 2019), as a potential origin of natural

hazards (e.g. Carrivick and Tweed, 2016), and to project their sea-level rise contribution (e.g. Marzeion et al., 2020). To monitor the evolution of a glacier and its response to climatic conditions, the mass balance is one of the most relevant variables (Haeberli and Hoelzle, 1995; Kaser et al., 2006). Based on in situ observations, the mass balance components can be accurately monitored at high temporal and spatial resolution (Pelto, 2000; Braithwaite, 2002; Thibert et al., 2018). Seasonal or annual





mass balance is measured with the glaciological method, i.e. ablation stakes and snow pits distributed across the glacier surface

(Cogley et al., 2011; Zemp et al., 2013; Sold et al., 2016; McGrath et al., 2018; Andreassen et al., 2020). Although these measurements require access to the glacier, they can be performed by relatively limited and simple technical equipment (Østrem and Stanley, 1969). This is also the reason why some point observations have already been performed since the very beginning of glaciological research in the late 19th century based on consistent methodology (Mercanton, 1916). Similarly, some series have been continued until today with few or no interruptions (e.g. Holmlund et al., 2005; Thibert et al., 2008; Huss et al., 2021),

especially in the northern hemisphere. Worldwide, several hundred glaciers have been monitored with in situ measurements but less than 50 were observed for periods longer than three decades (Zemp et al., 2009). The World Glacier Monitoring Service (WGMS) collects this data from contributors over all continents, thus merging national datasets into a global database.

Glacier mass balance represents an Essential Climate Variable (ECV, Bojinski et al., 2014), and is one of the most important variables of glacier and climate monitoring (Zemp et al., 2015; Trewin et al., 2021). However, the mass change of a glacier

cannot be directly measured in the field and thus relies on the spatial extrapolation of a discrete number of point observations to the entire glacier area (Cogley et al., 2011). Glacier-wide mass balance is thus never a direct observation, but a quantity inferred by extrapolation methods of varying complexity based on point mass balance measurements. Hence, these point measurements are at the basis of glacier-wide mass balance estimates. In addition, they are also highly valuable to enhance our understanding of the glacier-climate linkage (Vincent et al., 2017).

For the Swiss Alps, the national programme Glacier Monitoring Switzerland (GLAMOS) collects, compiles and makes available point mass balance measurements, among other variables, with a high spatio-temporal coverage. The first observations date back to the end of the 19th century and some glaciers have been continuously monitored for over 100 years (GLAMOS, 1881-2020). Furthermore, direct mass balance observations covering more than three decades are available for 20 glaciers, thus offering an exceptional spatial density of such long-term data series. This availability of in situ measurements is unique

worldwide and provides a great chance to further study the effects of climate forcing on glaciers over a centennial period.

The variety of primary sources from which these individual mass balance measurements are obtained, however, holds a number of challenges when aiming at compiling a high-quality dataset with fully documented metadata. For example, snow/ice density that is needed for converting a layer thickness change to a mass change can not be measured at every location in the field. Assumptions are thus needed but, so far, no clear indication on their basis was available. As previous observations have been

acquired within projects with widely differing scientific objectives, the processing levels and data formats can strongly diverge. In addition, some of the historic data had only been partially digitized, and in most cases a systematic, well-documented and complete collection of the data was missing. Similarly, some data sources had not yet been incorporated at all as they were unknown, or not deemed important. To provide homogeneous datasets with quantified uncertainties, it is crucial to obtain a better understanding of the origin of the glacier mass balance data. Given the centennial time frame of data acquisition in

Switzerland, as well as the many old and very diverse documents on which these observations have been conveyed to present times, it is important to rescue this raw data now, and to safeguard it for future investigations.

This study presents an updated and complete dataset of all historic point mass balance measurements on glaciers in the Swiss Alps including metadata. The dataset can be separated into measurements for an approximately annual period, the





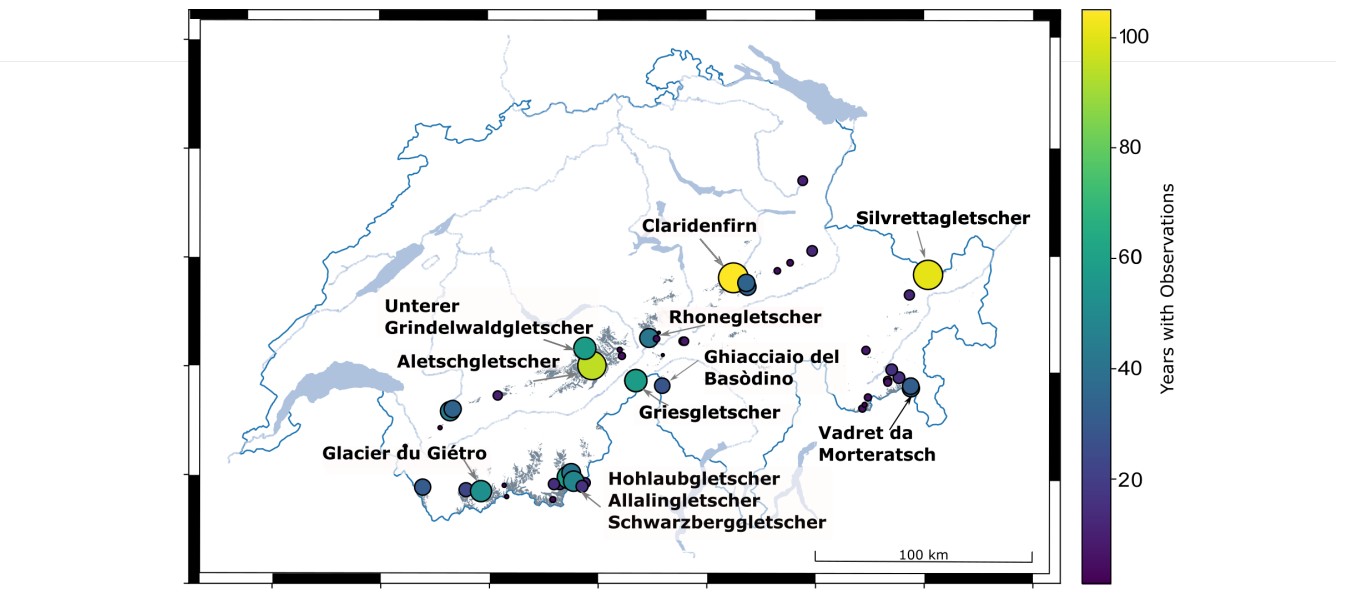

**Figure 1.** Location of Swiss glaciers with point mass balance measurements. The size and color of each dot indicates the number of years with measurements. Glaciers with long time series are labelled.

winter season, and shorter time intervals (days to months). The newly developed format includes metadata that capture, among

others, the measurement technique, the completeness, the quality of the mass balance readings, as well as the source and/or the

observer. To ensure consistency within the very heterogeneous data sources, they were systematically identified and revisited.

Existing entries were verified or corrected, and new data points as well as metadata were added. All data entries were checked

for homogeneity, and missing information, such as density information or position, was added based on consistent procedures.

The attributes in the metadata were then used to derive an estimate of the uncertainty of each individual point observation. This

effort resulted in data series supplemented with metadata for 63 individual glaciers throughout Switzerland, corresponding to

more than 60'000 point observations over almost 140 years. One third of this data was not available in digital format before

this study.

## 2   Study sites, data types and acquisition methods

### 2.1   Study sites and observations

The abundance of in situ glacier observations in the Swiss Alps is related to the favourable accessibility due to good infrastruc-

ture with roads, cable cars and railways that has been established since the 19th century. This situation allowed for the creation

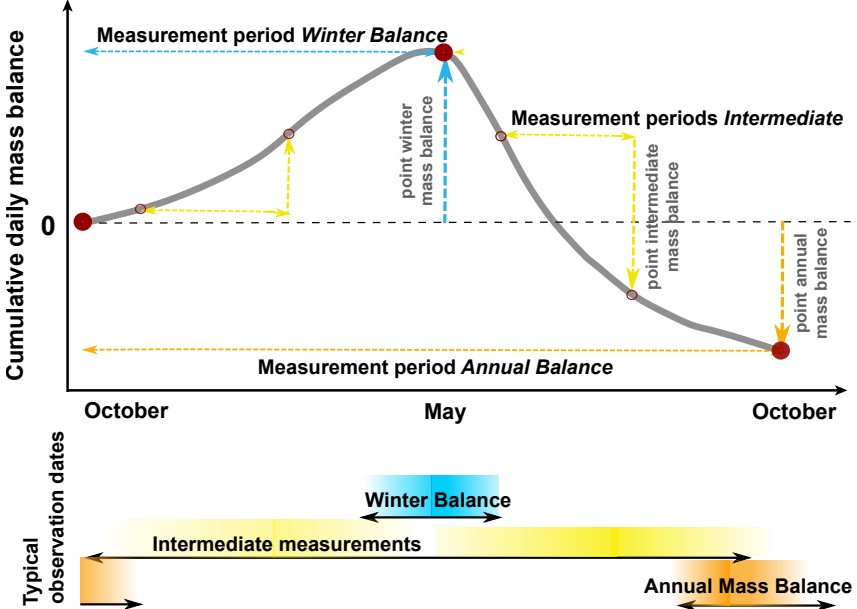

**Figure 2.** Schematic overview of time periods covered by point mass balance measurements. Typical observation periods for the annual measurements are from September to September, and from September to April/May for the winter period. Intermediate measurements refer to any sub-period throughout the year with an arbitrary length. Most of these measurements have been acquired over the summer season.

of an extensive glacier monitoring. The earliest direct mass balance measurements at a network of ablation stakes date back to 1884. In the meanwhile, mass balance measurements have been performed on more than 60 individual glaciers. Although most of the resulting time series are rather short, 20 glaciers have been monitored for at least three decades (Fig. 1). The longest

continuous time series covers more than a century without interruption (Claridenfirn, observations started in 1914), which is unrivalled worldwide. Similarly long but not quite as complete time series are also available for Silvrettagletscher (since 1915) and Grosser Aletschgletscher (since 1918). Other glaciers with observational series spanning more than half a century are located in southwestern Switzerland including Griesgletscher (since 1961), Allalin-, Schwarzberg- and Hohlaubgletscher (since 1955), and Glacier du Giétro (since 1967). Also observations on Ghiacciaio del Basòdino (since 1991), southern Switzerland,

have a coverage of 30 years. A more detailed description of the individual time series is presented in Section 4.1.

## 2.2  Measuring intervals

In situ point mass balance measurements refer to the observation of an added or removed layer of snow/ice at a given site over a defined period in time (Cogley et al., 2011). When combined with a measurement or an estimate of the layer's density, the mass balance in water equivalent (w.e.) can be determined. In this study, we divide all point observations into three categories

depending on the time interval covered. Our aggregation relates to the determination of (i) annual, and (ii) seasonal (win-



ter/summer) mass balance that is typically reported in international glacier monitoring (Zemp et al., 2015), but also features (iii) a class for shorter observation periods deviating from the classical timing of the field surveys.

We refer to *annual* measurements when the observations cover approximately one hydrological year, i.e. have been acquired between about late August and October of subsequent years (Fig. 2). The annual observations are either a mass gain in the accumulation area, or a loss in the ablation area. *Winter* measurements cover a period between the autumn minimum of glacier mass in the previous year and close to the spring maximum of snow cover on glaciers, which typically occurs between April and early June in the Swiss Alps (GLAMOS, 1881-2020; Gugerli et al., 2019). Finally, *intermediate* measurements refer to any shorter sub-period with a length of one day up to several months (Fig. 2). These observations provide insights into shorter-term glacier mass changes that might be valuable for mass balance model calibration and validation but have so far not been extensively exploited in seasonal glacier monitoring.

Every point mass balance observation is characterized by a start and an end date (Fig. 2, Cogley et al., 2011). In the ideal case, both dates are documented and known. Especially for historic measurements, however, this information has not always been reported precisely. For some measurement techniques, mainly applied for the winter period, the exact start date of the observation is unknown by definition. This is the case, for example, when detecting the previous year's (hardened) summer surface by snow probing or ground-penetrating radar (see Section 2.3).

### 2.3 Measurement techniques

### 2.3.1 Mass balance stakes

The most common and traditional method to observe point glacier mass balance is to place a stake in the ice/firn, usually at the end of the hydrological year, and to record its height above the surface (Østrem and Stanley, 1969; Kaser et al., 2003). By re-measuring this height after an arbitrary period of time, the thickness of the added/removed layer can be determined. If the density of the corresponding layer is known (from measurements or estimates), the mass balance can be computed. This method is applicable if the stakes can be regularly visited and maintained, at least at an annual interval. In Switzerland, aluminum or plastic poles are used at present. In the past, wooden stakes were common. Since this method is technically simple, it has been employed since the earliest measurements in the late 19th century, and the procedure has remained the same until today, thus allowing a high consistency in the observational series. Measurements at stakes can be used both in the ablation and accumulation area, and for all time periods (daily to annual). Results are most reliable at sites dominated by ablation, though, as stakes may sink into the firn in the accumulation area (Østrem and Stanley, 1969). Stakes placed in the accumulation area are thus primarily used to detect marked horizons identifying the last year's surface layer (Fig. 3A). Per glacier, between a single and up to 80 stakes were annually surveyed in Switzerland.

### 2.3.2 Automatic ablation stakes

Although the process of taking readings at mass balance stakes is straightforward, field access is required thus limiting the time resolution. Techniques for automatically acquiring more frequent measurements have recently been developed, such as

**Figure 3.** Illustration of different methods of point mass balance data acquisition: (A) Mass balance stake and marked horizon (accumulation area), (B) ablation stake with markers and automated camera-based readings, (C) snow probing, (D) ground-penetrating radar for measuring snow layer thickness, (E) snow pit excavation, (F) snow density measurement in a snow pit, (G) coring for snow density measurement, (H) historic method of nivometers (markers painted on vertical, ice-marginal rocks). Photos: M. Huss (A,E-G), C. Ogier (B-D), P.-L. Mercanton (H).

sonic rangers (Fitzpatrick et al., 2017), cosmic-ray sensors directly measuring snow water equivalent (Gugerli et al., 2019),
steel wires for continuous ablation measurements, or automated cameras monitoring ablation stakes (Landmann et al., 2021)
(Fig. 3B). Given the maintenance and cost, only one to very few such installations are feasible per glacier.

### 2.3.3 Snow probing

Snow probing is often used to measure the depth of the winter snow layer by detecting the ice surface or the harder horizon
of the last year's summer surface in the accumulation area (e.g. Sold et al., 2016; Pulwicki et al., 2018). As this method is
efficient, a few dozen up to several hundreds measurements spread over the entire glacier surface can be acquired by a single

team (most often consisting of two to three people) within a day. However, snow probing requires some experience and may result in erroneously detected horizons within the snow pack, such as ice lenses or an unclearly developed summer horizon in the firn area (Fig. 3C).

### 2.3.4 Ground-penetrating radar

In the last two decades, the use of Ground-Penetrating Radar (GPR), both helicopter-borne and ground-based, to detect winter
snow depth or firn layer thickness has gained substantial attention in glaciology (e.g. Dunse et al., 2009; Helfricht et al., 2014; McGrath et al., 2015). The technique was frequently used in the Swiss Alps as well (Machguth et al., 2006; Sold et al., 2013; Bauder et al., 2018). The dielectric contrast between snow and firn/ice, particularly in the ablation area and in dry snow, permits a rapid and continuous detection of snow depth (Fig. 3D). However, radar-wave velocity needs to either be estimated or be determined by dedicated measurements, and the processing of radar data can be ambiguous, especially in the accumulation
area (Sold et al., 2015).

### 2.3.5 Snow density measurements

To derive mass balance from the observed thickness of an added/removed layer, the layer's density needs to be measured or estimated. Whereas for bare-ice ablation typically a density of $900\,\mathrm{kg\,m^{-3}}$ is used (Kaser et al., 2003; Cuffey and Paterson, 2010), snow and firn density on glaciers can range from below $100\,\mathrm{kg\,m^{-3}}$ for fresh snow (Jonas et al., 2009) up to $800\,\mathrm{kg\,m^{-3}}$
for compacted old firn (Keenan et al., 2021; Ochwat et al., 2021).

 The density of winter or end-of-summer snow can be measured in a snow pit dug down to a marked or clearly identifiable horizon (Østrem and Stanley, 1969). The density is then calculated by measuring the weight of a snow tube with a known volume (Fig. 3E/F). This allows resolving the variation in density throughout the snow pack in addition to the bulk density that is required to infer local mass balance. Especially for high snow depth or strongly compacted end-of-summer snow, this
method can however be laborious and time-consuming. Alternatively, snow density on glaciers is also often determined by coring (Fig. 3G), whereby the same principle (weighting the snow mass of a core with known volume) is used.

### 2.3.6 Nivometers

Between about 1910 and 1950, so-called nivometers have been used at about half a dozen sites throughout Switzerland to monitor height changes at the glacier margin. Nivometers are scale marks that are painted on vertical rock faces next to the
glacier surface (often in the accumulation area, Fig. 3H). These sites were regularly visited by alpinists or the guardians of nearby mountain huts, and readings of the relative surface height were taken. Strictly speaking, nivometers do not provide a measure of mass balance since the result is also influenced by ice flow and snow/firn compaction. Furthermore, the location close to the glacier's edge might favour particular, non-representative accumulation/melt rates (e.g. Helfricht et al., 2015). Hence, the nivometers exhibit a much higher uncertainty than other methods and require careful interpretation. However, they





still provide valuable information about localized variations in accumulation and ablation at relatively high temporal resolution (often up to monthly), and in a time period with no or very limited other direct observational evidence.

### 2.3.7  Other techniques

Some data points refer to less conventional or accurate methods to monitor local glacier mass balance. For example, from the 1950s to the early 1990s annual accumulation layers visible in seracs or cornices were detected and measured by telescope on

several glaciers, sometimes at monthly resolution. This observation has considerable uncertainties though. At one site, a series of ten annual accumulation layers and corresponding densities from a firn core were incorporated in the data set. Very few entries also describe the position of the end-of-summer snowline, indicating a balanced local mass budget.

### 2.4  Measurement locations

Although the change of mass at a point is the most important observation, the location of the measurement site also needs

to be reported. In addition to revealing the average flow speed of that point over the respective time interval, the position is required for further interpretation of the observed mass balance, for example related to local terrain indices (elevation, slope, aspect etc.), or for spatially extrapolating mass balance to the entire glacier. Most of the earliest measurements did not include an exact record of the point location, and the position needed to be reconstructed. If the measurement site and its location was maintained over a long time period, this was possible based on the first accurately known coordinate, determined e.g. by

theodolite. In a few cases, however, only qualitative descriptions of the location, or a rough estimate of the measurement's elevation, were available. The location of the measurement sites is known with a higher accuracy after roughly the 1960s, for example by triangulation with theodolites. After about 2000, the use of handheld or differential GPSs is common.

### 2.5  Data sources

The data sources, scientific contexts and formats in which mass balance measurements on Swiss glaciers have been performed,

are highly diverse. The sources can generally be classified into three categories: primary, secondary and tertiary data.

Primary data is acquired directly in the field, often in field books. Whereas some original field records from measurements in the 20th century are still available in archives today, working with these sources is tedious due to handwritten text and a lack of context and structure in the notes. In more recent years, primary data were also digitally acquired. These include, for example, digital records of the field observations, the raw data of GPR surveys, or automated ablation measurements.

Secondary sources are typically tables in which those primary data were assembled. A processing in a basic way is already available in these sources (e.g. height differences were computed from raw stake readings). In this project, the majority of data was taken from secondary sources since they offer a readable format but are still very close to the original, unprocessed data. For historic sources, these tables are often handwritten on printed templates to ensure a consistent format. For more recent measurements, secondary sources are usually digital. However, the digital formats vary strongly, making automated processing

difficult, especially when no or incomplete documentation of the formats is available.



Tertiary sources are published works in which these measurements have been incorporated, e.g. a project report, a thesis, or a scientific publication. Whereas many tertiary sources focus on one specific glacier (e.g. Mercanton, 1916; Müller and Kappenberger, 1991) or one region (e.g. Huss and Bauder, 2009), there are also reports covering the whole of the Swiss Alps summarizing the work from different projects, such as the series of annual glaciological reports starting 140 years ago (GLAMOS, 1881-2020). These publications were originally in French, later in French and German, and since 1999 in English. From 1914 to 1978, the series "Firnberichte" focused on the observation of snow accumulation in the accumulation area of selected glaciers (Firnberichte, 1914–1978). Before the 1950s, all point mass balance measurements were individually included in these publication series, making them a valuable resource for a period in which barely any primary or secondary sources are available anymore (e.g. Firnberichte, 1914–1978; GLAMOS, 1881-2020). A substantial share of the data refers to projects related to hydropower dam constructions (e.g. Griesgletscher, Mattmark, Mauvoisin) or, later, stem from research projects with a variety of scientific objectives (e.g. Bauder et al., 2003; Farinotti et al., 2010).

Apart from this large collection of historic measurements on glaciers that have already been the focus of previous studies (e.g. Huss et al., 2015), also unusual and/or unexpected sources were discovered in the frame of the data rescue action. For example, detailed data for four small glaciers in Eastern Switzerland (Blau Schnee, Jörigletscher, Glatschiu dil Segnas, Glatscher dil Vorab) covering the last decade were collected by O. Langenegger, a retired private person. Thanks to the detailed documentation, this data was successfully integrated, thus adding to the diversity of the dataset.

## 3 Methods

### 3.1 Dataset formatting

To account for the great variety of sources and types of point mass balance measurements, a new data format was developed. Besides the actual mass balance observation, information on the date of the surveys, the position and elevation of the measurement, the density, as well as a code for the source/the observer allowing traceability of each entry is provided. Additional relevant meta-information on the individual data points is encoded in a series of attributes that are efficiently assigned to the thousands of entries and are machine-readable (Table 1). Additional information is defined in five attributes: (1) The 'date quality' allows assigning whether the observation dates are known, or if they have been estimated. (2) The 'position quality' describes if and how the location was measured, or if it was estimated. (3) The 'density quality' describes if the snow density was directly measured at the site or was estimated, and how this estimation was performed. (4) The 'measurement type' denotes the measurement technique (e.g. stake reading, snow probing). (5) The 'measurement quality' characterizes a relative accuracy level of each measurement compared to entries of the same type, thus allowing the specification of potential problems (e.g. if an ablation stake was not visible due to fresh snow but it was still possible to reconstruct a value). A detailed description of all attributes is provided in Table 1. The combination of attributes, and an average uncertainty assigned to individual components of point mass balance, allows an integrative uncertainty estimate for each observation (see Section 3.3).





**Table 1.** Attributes of the metadata. All elements of the five attributes are described including typical uncertainties. Note that the uncertainty for unknown observation dates (14 days) has been translated into a corresponding mass balance error depending on elevation. See Section 3.3 for details on the procedures to assign the uncertainties.

|  | Assigned uncertainty | Description |
|---|---|---|
| | **date quality** | |
| 0 | ±0.15-0.35 m w.e. | Start and end dates estimated/unknown |
| 1 | ±0 m w.e. | Start and end dates exactly known |
| 2 | ±0.11-0.25 m w.e. | Start date exactly known, end date estimated/unknown |
| 3 | ±0.11-0.25 m w.e. | Start date estimated/unknown, end date exactly known |
| | **position quality** | |
| 0 | ±100 m | Unknown |
| 1 | ±0 m | Measured by differential GPS |
| 2 | ±5 m | Measured by handheld GPS |
| 3 | ±0 m | Measured using an alternative method (e.g. theodolite) |
| 4 | ±20 m | Estimated from previous measurements |
| 5 | ±200 m | Estimated based on elevation information |
| | **density quality** | |
| 0 | ±12% | Quality/source unknown |
| 1 | ±1.5% | Ice density |
| 2 | ±5% | Measured snow density |
| 3 | ±8% | Density of snow estimated from nearby measurements |
| 4 | ±12% | Density of snow estimated without nearby measurements |
| 5 | ±10% | Only water equivalent information (density and source unknown) |
| 6 | ±12% | Estimated based on linear regression in post-processing (Eq. 1) |
| | **measurement type** | |
| 0 | ±10 cm | Unknown |
| 1 | ±5 cm | Mass balance stake |
| 2 | ±10 cm | Snow probing / snow pit / coring |
| 3 | ±5 cm | Marked horizon (e.g. for snow pit or coring) |
| 4 | ±5% of snow depth, min. 10 cm | Ground-penetrating radar |
| 6 | ±150 cm | Nivometer |
| 7 | ±1.5 cm | Automatic real-time cameras |
| | **measurement quality** | |
| 0 | ±30 cm | Quality/source unknown |
| 1 | ±0 cm | Typical uncertainty for specific method (see measurement type) |
| 2 | ±20 cm | High reading uncertainty (e.g. stake bent) |
| 3 | ±40 cm | Reconstructed value (e.g. stake melted-out, stake buried by snow) |





## 3.2 Data gaps

Although the historic point mass balance measurements were generally reported at a high level of completeness, some observations lack certain metadata. In most cases, this lack is related to the systematic scarcity in snow density measurements due to their laborious acquisition. Sometimes also the exact date of the observation or the elevation are not documented. Therefore, it was necessary to consistently fill these data gaps. This was achieved by a statistical evaluation of all direct observations and subsequent extrapolation to missing entries. All estimated values remain fully traceable as they are flagged via the corresponding attributes.

### 3.2.1 Estimation of missing observation date

In case the start or end date, or both dates, were unknown for the annual measurements, the average date at which the observations were performed at the respective glacier was used and supplemented in the dataset. This procedure accounts for the fact that the typical time frame for annual measurements can vary by up to one month between glaciers, but often the same groups of glaciers are visited in a similar order each year. Therefore, this approach is likely to be more accurate than using the same date across all glaciers. For the unknown starting date of winter accumulation measurements by snow probing or GPR, no date was supplemented in the dataset as the date of the reference horizon differs for every location, typically being later in the season in the ablation area compared to the accumulation area. The corresponding date could be reconstructed retrospectively based on mass balance modelling detecting the local date of last summer's minimum horizon. This is not included in this observational data set, and a higher uncertainty is attributed to these observations (see Section 3.3).

### 3.2.2 Estimation of missing elevation

In some cases, only the coordinate of the measurement site was recorded but not its elevation, which was retrieved from corresponding digital elevation models (DEMs), if possible for a similar point in time. For large Swiss glaciers, DEMs are available at intervals of one to a few decades back to the early 20th century based on aerial photogrammetry or topographical maps (Bauder et al., 2007). For the remaining glaciers, products from the Federal Office of Topography (swisstopo) covering the entire Swiss Alps were used. The DHM25 with a spatial resolution of 25 m from around 1980 (Fischer et al., 2015) was employed for earlier observations, and the most recent release of the SwissAlti3D DEM (swisstopo, 2018) at 2 m resolution (originating from between 2013 and 2018) was used after 2000.

### 3.2.3 Estimation of missing snow density

A direct determination of density of the added/removed layer is not feasible for every point observation for practical reasons. Often such a measurement is only performed at one to few sites per glacier during each survey. Consistently estimating the density for unmeasured locations is thus crucial. For ice ablation, the density can be assumed to be rather homogeneous and has been set to $900\,\mathrm{kg\,m^{-3}}$ as recommended previously (Østrem and Stanley, 1969; Kaser et al., 2003). In extreme years and at a few sites, ablation can also occur over firn layers, i.e. when the snowline rises into an elevation where typically





accumulation prevails. Retrospectively determining the average density of the lost one- to multi-year firn is impossible, and a density of between $600\,\mathrm{kg\,m^{-3}}$ and $800\,\mathrm{kg\,m^{-3}}$ has been assigned depending on the approximate age of the removed firn layer.

A detailed estimation of the density of winter snow, as well as end-of-summer snow is important as a vast amount of data point is affected, and snow density can vary greatly in time and space (e.g. Bormann et al., 2013; López-Moreno et al., 2013; Sold et al., 2015; Senese et al., 2018).

A total of 815 direct snow density measurements with for a water equivalent of beyond 0.1 m w.e. (344 for the annual period, and 471 for the winter period) was analyzed. This corresponds to 13% of all annual point measurements experiencing

net accumulation, and to 2% of winter snow observations (excluding densely-spaced GPR profiles). We note that this only includes instances where it is unambiguously documented that snow density has been measured on site. More observations might have been performed that are no longer traceable from the original data sources. Although the coverage of in situ density measurements might seem relatively limited, this is probably one of the largest consistent compilations available for snow density on glaciers.

Multiple linear regression with the bulk density of the snow layer was performed for three variables: (i) the day of year (i.e. the timing of the observation), (ii) the elevation of the measurement point, and (iii) local snow depth (Fig. 4). For the winter period, a significant, yet relatively weak correlation was found with clear positive dependencies on the date of the measurement and snow depth (Fig. 4A-C). This is also supported by studies in non-glacierized terrain (e.g. Jonas et al., 2009) and is related to the aging of snow. The unclear dependency of winter snow density on elevation is likely explained by the counteracting

processes of compaction with higher snow depth and the occurrence of melting, and hence densification, at lower elevation during the time of the surveys in April/May. The multiple linear regression explains 40% of the variance. The relation with the day of year $t$ (days), the elevation $z$ (m) and snow depth $d$ (cm) allowed estimating missing densities $\rho$ ($\mathrm{kg\,m^{-3}}$) for the winter period as

$$\rho = 258 + 1.22 \cdot t + 0.01 \cdot z + 0.14 \cdot d. \tag{1}$$

No dependency of observed snow density on any variable was found, however, for the annual period according to end-of-summer measurements (Fig. 4D-F). We interpret this with other processes, such as the share of fresh snow at the date of observations in September/October or the amount of summer melting. These factors appear to be more relevant and override the signal of expected snow densification over an entire year. We therefore use the overall average of all end-of-summer snow density observations ($539\,\mathrm{kg\,m^{-3}}$) to supplement a density estimate for missing entries at the annual scale.

## 3.3 Uncertainty

The uncertainty in direct mass balance measurements can vary greatly depending on the measurement technique and the assumptions involved. Hence, accounting for the uncertainty of each point observation is highly important to obtain a useful dataset for further applications, such as the evaluation of glacier-wide mass balance or mass balance model calibration and validation. Our new dataset including complete metadata in addition to the actual measurement values allows estimating the

accuracy of each individual data point. This is achieved by assigning a typical uncertainty to the different scenarios of data

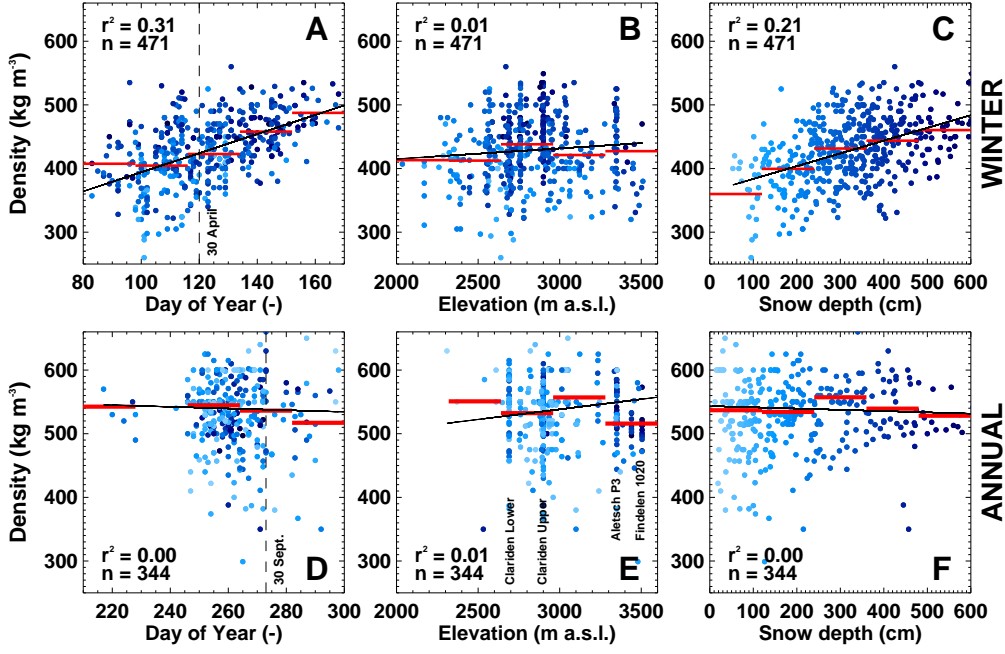

**Figure 4.** Dependency of measured snow density for (A-C) the winter season, and (D-F) the annual period on the day of year of the observation, the site's elevation and, local snow depth. The shading of the dots qualitatively indicates low (light) to high (dark) snow depth. The linear correlation coefficient $r^2$ is given for each variable, and the number $n$ of measurements is stated. The straight line corresponds to a linear fit, and bold red lines show average densities for five classes of the analyzed variable for illustration. In (E) four sites with many repeated density observations are highlighted, explaining the vertical arrangement of the dots.

acquisition that are in turn described by a combination of attributes documenting the individual elements of point mass balance determination.

For each entry of the five attributes listed in Table 1, a generic uncertainty contribution was determined based on (i) statistical evaluation of the dataset, if applicable, (ii) literature, or (iii) qualitative knowledge from field experience. We acknowledge that
the assigned uncertainties represent averages, or rough estimates, and might thus not be appropriate for all cases. Nevertheless, we deem them valuable as they allow a consistent application to the whole dataset.

An unknown measurement date (date quality, Table 1) cannot be directly translated into an uncertainty in mass balance. We therefore first determined the standard deviation of all reported measurement dates ($\pm 14$ days), which we assumed to represent the potential error in the date estimate when it is not documented. Then, we evaluated typical changes in mass balance around
the average date of the observations during the time when these dates are unknown (almost exclusively in September) from our dataset of intermediate point mass balance measurements. We find a clear dependence of bi-weekly local ablation rates on elevation, with stronger melting within the uncertain dating of measurements at low elevation (not shown). Higher uncertainties were thus assigned to sites at low elevation and lower uncertainties to sites at high elevation. Uncertainties in the x/y-coordinates of the sites (position quality) were assigned but were not further translated into an uncertainty in point balance. This is because





no systematic dependencies could be found. For the measurement type, typical reading uncertainties were assigned based on either direct field experience of active observers or on values given in the literature (e.g. Huss and Bauder, 2009; Zemp et al., 2013; Beedle et al., 2014). Values range between 1.5 cm for automatic stake readings (Landmann et al., 2021), 10 cm for snow probing, to up to 150 cm for nivometers (Table 1). For GPR-based snow depth measurements, a 5% uncertainty relative to snow depth was assigned beyond a minimum uncertainty of 10 cm to account for the fact that the wave propagation speed is

most often estimated, rather than determined in the field. The measurement quality attribute is used to account for additional uncertainties that arise when using data with unknown origin, or to accommodate special situations (e.g. the reported bending of an ablation stake). The assigned uncertainties are based on field experience (Table 1). The individual contributions from the various components mentioned so far are then combined with the root sum-of-squares, and are translated into a mass balance reading uncertainty $\sigma_{\mathrm{reading}}$, stated in m water equivalent, after multiplying with density.

The uncertainty in the density information $\sigma_{\mathrm{density}}$ (density quality, Table 1) is specified relative to mass balance, and varies from 1.5% for bare-ice ablation to 12% when the density is estimated from the statistical relation in Eq. 1, or without any supplementary information. Uncertainties have been either assigned based on literature (Cuffey and Paterson, 2010; Sold et al., 2016; Gugerli et al., 2019; Huss et al., 2021), the observed standard deviation in measured densities (Fig. 4), or multiple density measurements in the same snow pit. We finally combine the reading uncertainty and the density uncertainty into a point mass

balance uncertainty $\sigma_{\mathrm{point}}$ as

$$\sigma_{\mathrm{point}} = \sqrt{\sigma_{\mathrm{reading}}^2 + \sigma_{\mathrm{density}}^2 \cdot b}, \qquad (2)$$

where $b$ is the point mass balance. Typical uncertainties in point balance measurements and their dependency on time periods and measurement techniques are discussed in Section 4.4.

## 4   Results and Discussion

We first present the spatio-temporal coverage of the new point mass balance dataset. Afterwards, selected aspects of the dataset are shown. We discuss the changes in point mass balance over the last century, both at the seasonal and the annual scale. Furthermore, we investigate the long-term changes in observed snow density, and discuss the contribution of methodological approaches to the overall uncertainty in point mass balance measurements.

### 4.1   Extent of new point mass balance dataset

A complete and documented dataset comprising all currently accessible point mass balance measurements that were acquired on Swiss Glaciers since 1884 has been compiled. This results in entries for 63 individual glaciers with 62'304 data points. Thereof, annual data with 10'433 observations are available for 55 different glaciers, 39'873 point measurements for the winter period are available distributed over 44 glaciers, and data for intermediate periods are available for 46 glaciers, featuring 11'998 data points. For roughly two thirds of the point observations, existing digital datasets compiled and digitized in previous projects

could be used. If the original sources of this data (primary, secondary, tertiary sources) were accessible, they were revisited,



missing information was complemented or corrected if necessary, and metadata for each entry was added. About 20'000 new mass balance point observations that were so far not digitally available, inaccessible, or simply unknown, were added as well. Most of these additions refer to the intermediate measurements, as this data type had not been systematically integrated into scientific studies so far. Also about two dozen additional, mostly short series of seasonal/annual mass balance were included

in the dataset. These refer to early monitoring efforts in the beginning of the 20th century, as well as the past two decades. For most of the existing digital time series, revisiting the source data resulted in the inclusion of additional point measurements, or in the documentation of important metainformation (e.g. start/end dates of observations, density measurements). The sources of this newly added information strongly vary. In some cases, original field books could be accessed (primary data, e.g. Aletsch, Silvretta). Mostly, information was retrieved from compilations of measurements in original handwritten, printed or – for newer

measurements – digital tables (secondary data, e.g. Allalin, Gries, Jöri). Many point data were also retrieved from historic publications (e.g. early volumes of the series GLAMOS, 1881-2020), where individual measurements were partly documented in great detail (tertiary data, e.g. Orny).

The number of simultaneously observed glaciers regarding annual mass balance has increased from one in the 1880s to 30 in 2014 (Fig. 5A). Between about 1910 and 1950, 5-10 glaciers were monitored, while the number increased to 10-15 in the

decades afterwards. Throughout the last two decades the number of glaciers with annual point balance measurements strongly increased. Although more glaciers are observed now in comparison to the 1960s and 1970s, the number of individual point observations was at a maximum at that time: More than twice as many annual measurements (18.3 points per glacier and year) were performed, whilst that number has dropped to 7.2 on average since 2000 (Fig. 5A). Monitoring is thus less focused on few sites today but attempts to better cover the local and regional differences in mass balance with a reduced effort per glacier.

Throughout the last century, winter accumulation was observed on a smaller number of glaciers compared to annual balance, with surveyed glaciers ranging between 3 and 14 before 2010 (Fig. 5B). In the last years, the monitoring of winter snow was included in additional monitoring programmes. In comparison to the annual surveys, the average number of measurement points per glacier has a completely different pattern: While only very few local winter snow measurements have been conducted throughout the 20th century (between 1.4 and 3.4 points per glacier and year on average), this number sharply increased in

the recent years to above 100 points per glacier and year. This is explained by a change in the strategy to monitor seasonal glacier mass balance: Until about 2005, snow was just measured at a few index locations per glacier not aiming at resolving the glacier-wide winter balance. During the last decades, however, the efficient observation of spatial patterns in winter snow distribution by probing and the application of GPR has taken over providing a dense spacing of measurements.

The length and completeness of the datasets for the individual glaciers strongly vary (Fig. 6). In the following, some examples

of outstanding or otherwise interesting series of point mass balance measurements – encompassing different classes from long and complete, to short and discontinuous – are shortly described. Resulting long-term changes are presented in Section 4.2.

Point mass balance measurements at two sites were acquired in seasonal resolution on Claridenfirn continuously since 1914, making it the longest and most complete direct mass balance record worldwide (Huss et al., 2021). The data also are exceptionally homogeneous, and were well documented in several dedicated publications (Müller and Kappenberger, 1991; GLAMOS).

Very long series of seasonal mass balance are also available for Silvrettagletscher and Grosser Aletschgletscher, where 1-2 sites





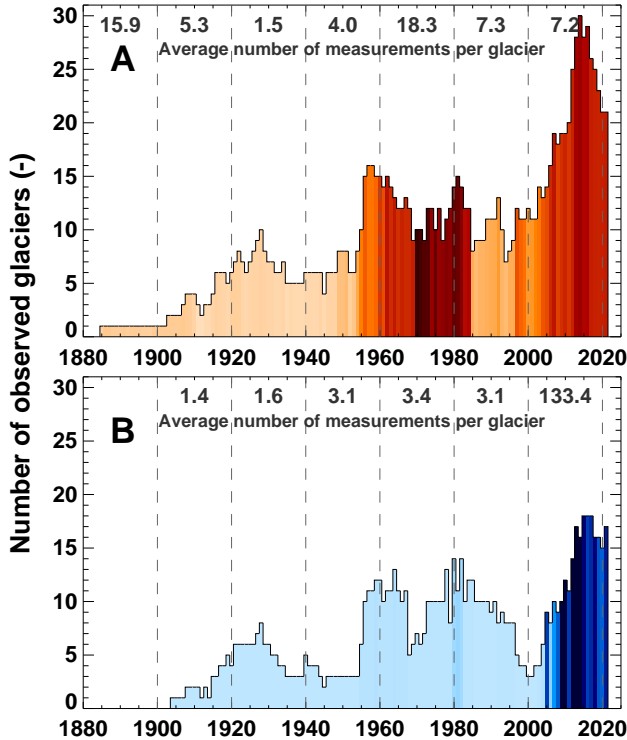

**Figure 5.** Number of observed glaciers over (A) the annual and (B) the winter period. The shading qualitatively indicates the total number of conducted point measurements per year, and a two-decadal average of point observations per glacier and year is given on top.

have been continuously monitored since 1915 and 1918, respectively, mostly seasonally (Huss and Bauder, 2009). Sometimes, readings with up to monthly resolution are available. At both glaciers also extensive monitoring programmes, with 40 and more stakes at times, were set up after the 1950s but were resized again in the mid-1980s (Fig. 6).

Long and continuous time series since the 1950s/60s are also available for several glaciers in the Canton of Valais (hydrolog-
ical basin of the Rhone), related to hydropower projects. Whereas the monitoring networks at e.g. Allalingletscher and Glacier du Giétro only consist of 5-10 stakes observed annually, up to 80 sites were surveyed on Griesgletscher until the mid-1980s (Huss et al., 2009a). Detailed seasonal monitoring programmes with dense networks of measurements were also started in the late 1940s on Limmern- and Plattalvafirn, northeastern Switzerland, but were unfortunately discontinued in the 1980s (Braun et al., 1994). Continuous seasonal monitoring at a dense stake network is performed since 1991 on Ghiacciaio del Basòdino,
southern Switzerland (Fig. 6). On Vadret Pers, southeastern Switzerland, a network of stakes in the ablation area has been observed since 20 years (Zekollari and Huybrechts, 2018).

The earliest systematic mass balance measurements on any glacier worldwide were performed on Rhonegletscher. From 1884 to 1909, a network of up to 20 mass balance stakes was maintained in the ablation area and data was excellently documented (Mercanton, 1916). This marked a milestone in glaciology. The detailed monitoring on Rhonegletscher has been
resumed only in 2006, after three years of with intermittent observations around 1980. In the first decades of the 20th century,





**Figure 6.** Overview of the temporal coverage of point mass balance series for the annual period (orange-red), the winter period (light-dark blue), and intermediate periods (light-dark green). Light colours qualitatively indicate that few data per year are available, and dark colours refer to a high number of point measurements. Glaciers are ordered according to the total length of the period covered by measurements (given in parenthesis). On the right, the identifier according to the latest Swiss glacier inventory (Linsbauer et al., 2021), as well as the corresponding area and elevation range is given for characterizing the glacier.





measurement sites have been set up in the accumulation areas of several glaciers for investigating high-mountain precipitation, e.g. at Glacier d'Orny, Glacier du Tsanfleuron (Valais), or Vadret d'Err (Eastern Switzerland). These early series where discontinued after a few decades but still provide important local insights into glacier mass balance variations, partly with up to monthly resolution, from a period with otherwise little direct information (Fig. 6). Some of the early series also consist of

observations with a high uncertainty (nivometers, telescopic detection of accumulation layers) and need to be interpreted with care (e.g. Unterer Grindelwaldgletscher, Vadret da Morteratsch).

During the last two decades, several additional, detailed programmes with seasonal monitoring at an extensive stake network have been set up both on large (e.g. Findelengletscher, Glacier de la Plaine Morte) and small (e.g. Pizolgletscher) glaciers (Fig. 6). This completes the spatial coverage throughout all regions of the Swiss Alps (Fig. 1). Local mass balance observations

with a coverage of up to 10 years were also included for four glaciers (e.g. Blau Schnee, Jörigletscher) for which no data were reported in the literature so far.

Furthermore, short and discontinuous series related to individual projects, conducted without the aim of long-term mass balance monitoring, were detected and included in the dataset (e.g. Gornergletscher, Glacier de Saleina, Gurschenfirn etc., Fig. 6). These observations are mostly byproducts of other research activities, sometimes just cover one or a few years, and

mostly consist of only a small number of measurement sites per glacier, thus precluding the computation of a glacier-wide mass balance. Nevertheless, we argue that including these datasets is important, both for promoting completeness and because the data would most likely be lost otherwise.

## 4.2 Long-term point mass balance series

A selection of some of the longest and/or most consistent series at individual points, both for the annual and the winter period,

is presented in Figure 7. While the point mass balance is a highly localized measure that reveals no immediate information about the entire glacier's mass change, analyzing long-term variations in point balance can provide insights into the response of surface accumulation and melt to climate change (Vincent et al., 2004; Vincent et al., 2017). Unlike glacier-wide mass balance – an evaluated and spatially extrapolated quantity typically used in international glacier monitoring (e.g. Zemp et al., 2015) – point mass balance series are unbiased by the dynamic change in glacier extent (advance/retreat) and, hence, only mirror the

climatic impact on surface mass balance, independent of changes in glacier geometry.

For sites at high elevation, such as the stake called "P3" on Grosser Aletschgletscher, the observed averages of annual and winter balance are similar, indicating that summer balance is close to zero and even positive in some years (Fig. 7). The uppermost part of the accumulation zone is not much affected by melting, and mass balance is dominated by snow precipitation. The data indicate that the amount of local accumulation in the Alps has not undergone major changes over the last 100 years.

This is also supported by long-term precipitation observations at lower elevation, which are available throughout Switzerland (Isotta et al., 2019). For sites in the accumulation zone closer to the equilibrium line, however, point mass balance is more strongly affected by an increase in melt rates while still indicating stable winter snow accumulation (e.g. Clariden Upper site). Some long-term measurement sites have also transited from the accumulation to the ablation area (e.g. Clariden Lower

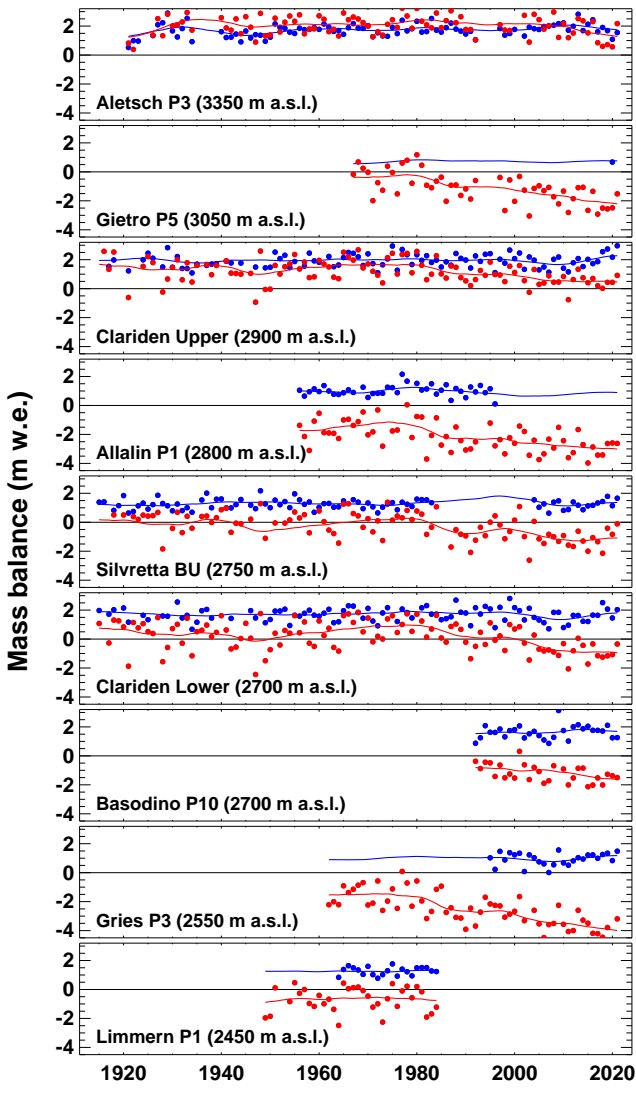

**Figure 7.** Evolution of the point mass balances observed at selected sites labelled with their original code (e.g. "P3" for Aletsch). Long and continuous time series at different elevation and throughout the Swiss Alps are shown. Red dots refer to observed annual mass balance, blue dots to winter balance. The lines show 11-year running means. For computing the running mean in this illustration data gaps were filled based on the approach by Huss et al. (2009b). The sites are ordered by decreasing elevation.

site, Silvretta "BU", Fig. 7) even though their absolute position was maintained over the years. Especially at these sites, the
decreasing trend in annual mass balance is evident.

For investigating the long-term changes in seasonal mass balance in more detail, we have selected the six longest and most complete point series. These correspond to the Swiss reference glaciers of the WGMS (Zemp et al., 2015). Direct comparability of the raw point measurements is hampered by the differing dates of the measurements. We thus first homogenize all



observations as to correspond to the period 1 Oct. to 30 Sept. (hydrological year) for the annual period, and 1 Oct. to 30 April
for the winter period. The homogenization follows the approach by Huss et al. (2009b). Thereby, a daily point accumulation
and melt model driven by nearby meteorological observations is constrained to exactly match both the winter and the annual
mass balance observation, and is then used to correct measured mass balance to a common period, as well as to fill occasional
data gaps by relying on average model parameters. In order to directly compare the series to each other, we define 1981-2010
as a reference period and compute the difference of each observation relative to the series' average over that time interval – an
approach commonly applied when analyzing point balance series (e.g. Vincent et al., 2017).

Figure 8 shows decadal anomalies in annual and winter balance, and underlines the more qualitative analysis of Figure 7.
It is evident that decadal variations in winter balance are much smaller than those of annual balance, hence indicating that
it is the summer season that drove changes in mass balance over the last 100 years. This finding is consistent with previous
studies (Vincent, 2002; Vincent et al., 2004; Huss et al., 2021). Although the long-term changes in winter balance are not fully
consistent between the analyzed series, a very weak increasing trend is apparent (Fig. 8A). This shows that the snow water
equivalent on glaciers on 30 April is not affected by rising air temperatures. In contrast, all selected series, except for the high-
elevation site on Grosser Aletschgletscher, show a substantial drop in local annual balance after 1980, accounting for about
–1 m w.e. (Clariden Upper site) to beyond –2 m w.e. (Gries "P3") until today. Sites in the ablation area are thereby experiencing
the most important changes (Fig. 8A). An evaluation of each series' standard deviation within the respective decade does not
reveal any consistent shift in the magnitude of year-to-year variability in mass balance, neither at the seasonal nor at the annual
scale (Fig. 8). At present, our dataset does thus not indicate an increase in extreme events relative to the respective decadal
mean.

## 4.3 Temporal changes in snow density

On average over all more than 800 direct observations in snow pits and by snow coring, the density of snow layers accumulated
over the annual period is $539\pm86\,\mathrm{kg\,m^{-3}}$, and the density of winter snow is $429\pm52\,\mathrm{kg\,m^{-3}}$. Although studies on the densi-
fication of firn layers over time are available (e.g. Ambach et al., 1989), the temporal evolution of snow density accumulated
at the surface of glaciers has never been assessed to our knowledge. This is due to the scarcity and the limited temporal scope
of such data. Our comprehensive dataset of snow density measurements on glaciers now allows such an analysis. Due to the
temporal and spatial discontinuity of the in situ data, as well as their uncertainties, we decided to not interpret annual values.
All available density observations between 1950 and 2020 were thus averaged in 10-year intervals, after temporally homoge-
nizing winter densities to 30 April using the densification rate of Equation 1. Density observations before 1950 are too scarce
and were not considered for this analysis.

Although in some decades the availability of direct density observations is limited, at least 14 individual measurements
(annual or winter) are available (Table 2). Between 1950 and present, no trends in decadal mean snow density at the glacier
surface statistically significant at the 95% level according to the Mann-Kendall test are visible, neither for the annual nor for
the winter period. A notably higher end-of-summer snow density (annual period) in the 1990s might be related to a different
subset of observed sites with different characteristics (Table 2). To exclude effects from shifts in the statistical sample we also



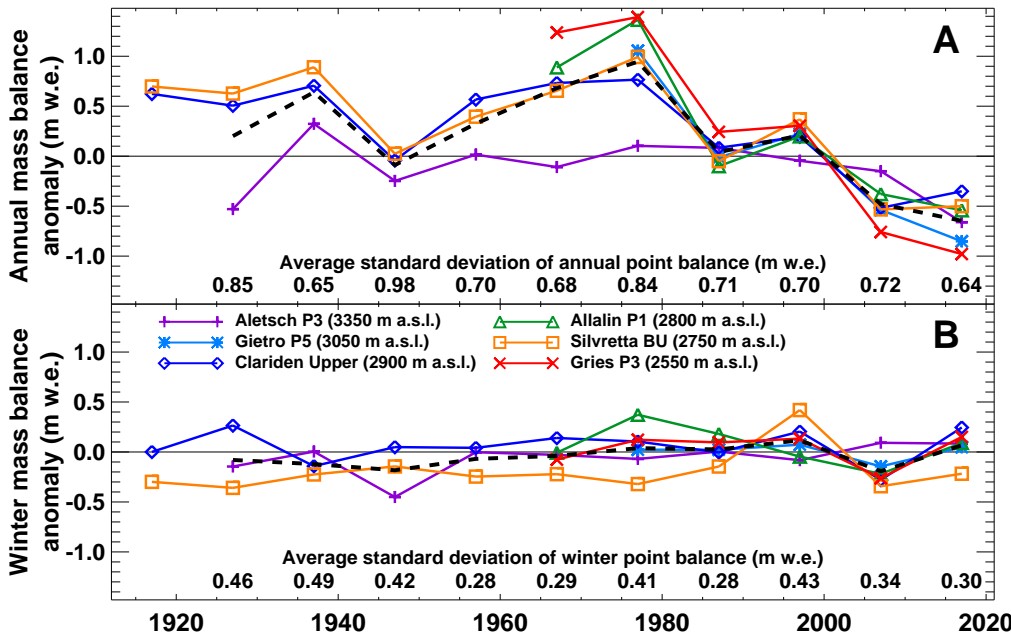

**Figure 8.** Anomaly in (A) annual and (B) winter balance relative to 1981-2010 observed at six selected sites with complete long-term observations, averaged over 10-year periods. The dashed line refers to the average anomaly of all selected sites. Note that the approach by Huss et al. (2009b) has been used to homogenise the point mass balance observations to common periods of the year (see text for details).

analyzed the most complete series of snow density measurements (both winter and annual period) on Grosser Aletschgletscher and Claridenfirn with partly over 60 years of continuous data. Also for these individual sites no statistically significant trends
in density were found. The absence of trends indicates that the accelerated mass loss of Alpine glaciers has not (yet) resulted in a detectable impact on the snow density on glaciers – neither for winter, nor the annual period – although this might be expected from higher winter temperatures and more intense summer melting.

### 4.4    Attribution of uncertainties

Computed uncertainties in point mass balance measurements (see Section 3.3) were aggregated into different classes to in-
vestigate their magnitude and relevance (Fig. 9). Typical uncertainties in annual mass balance measurements are found to be between 0.05 and 0.14 m w.e. (Fig. 9A). Absolute uncertainties in annual accumulation measurements are only slightly larger than for ablation. This is explained by the typically much higher magnitude of ablation compared to accumulation, resulting in a relative contribution of the uncertainty to measured point mass balance of about 3.4% for ablation, whereas it accounts for 12.9% for accumulation. Average uncertainties in winter mass balance measurements are higher, with 0.17-0.22 m w.e., or
16% of local mass balance on average (Fig. 9A). This is mainly explained by the undefined starting date for snow probing and GPR-based snow depth that refers to last fall's minimum surface. Using modelling, the corresponding date could be estimated though, thus reducing final uncertainties depending on the application.

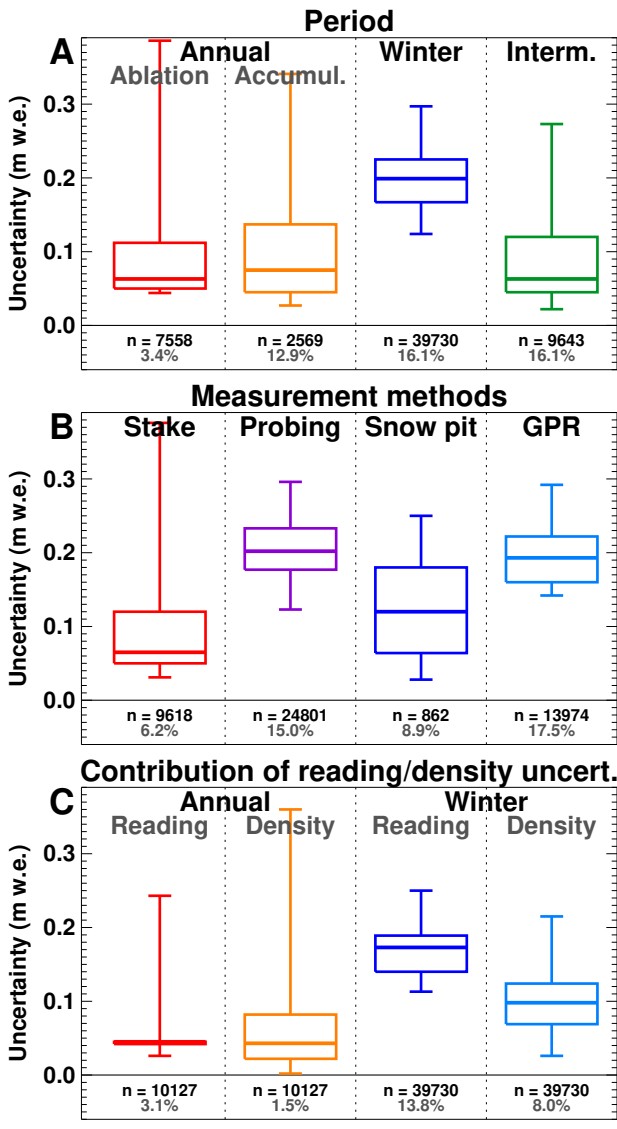

**Figure 9.** Uncertainty in point mass balance measurements aggregated into classes depending on (A) the time period covered, (B) the measurement method, and (C) a separate evaluation of the reading and density uncertainties ($\sigma_{\mathrm{reading}}$ and $\sigma_{\mathrm{density}}$, see Eq. 2), both for annual and winter balance. Note that in (A), ablation and accumulation are evaluated separately for the annual period. Boxes refer to the 25% and the 75% percentile, and bars encompass a 95% interval. The bold horizontal line shows the median. The number (n) of evaluated points is given, and the average share of the uncertainty relative to the point's mass balance is stated in grey.



**Table 2.** Dedacal averages of all observed snow densities both at the annual scale $\rho_{\mathrm{ann}}$ and for the winter period $\rho_{\mathrm{win}}$. $n$ is the number of documented direct measurements in the respective interval. The standard deviation of all measurements per period is given. Before averaging, values for the winter period have been homogenized to correspond to 30 April based on the densification rate of Eq. 1.

| Decade | $\rho_{\mathrm{ann}}$ (kg m$^{-3}$) | $n_{\mathrm{ann}}$ (-) | $\rho_{\mathrm{win}}$ (kg m$^{-3}$) | $n_{\mathrm{win}}$ (-) |
|---|---|---|---|---|
| 1951-1960 | 517±70 | 15 | 395±42 | 41 |
| 1961-1970 | 516±40 | 50 | 409±30 | 40 |
| 1971-1980 | 501±78 | 42 | 417±29 | 19 |
| 1981-1990 | 543±42 | 35 | 435±34 | 14 |
| 1991-2000 | 584±98 | 37 | 432±38 | 20 |
| 2001-2010 | 543±46 | 54 | 417±48 | 103 |
| 2011-2020 | 527±71 | 77 | 435±41 | 211 |
| Average | 533±69 | 310 | 424±43 | 448 |

Whereas absolute average uncertainties in mass balance measurements at stakes and in snow pits / from coring are smaller than 0.07 m w.e. and 0.12 m w.e., respectively, snow probing as well as snow depth measurements by GPR are more uncertain, with uncertainties in the order of 15-17% of the local mass balance on average (Fig. 9B). For both the annual and the winter measurements, the contribution of reading and density uncertainty to the total uncertainty is similar, with a tendency for the reading uncertainty to be more important for the winter period (Fig. 9C). The latter is again explained by the relatively high reading uncertainty assigned to snow probing and GPR, representing the bulk of winter observations in terms of the point number.

## 5 Conclusion

Point measurements are the original and raw observations of glacier surface mass balance. They thus represent the backbone of the in situ monitoring of glacier mass changes. Even though these observational series cover up to a century at some sites, relatively little effort had so far been invested in Switzerland to fully document all measurements, to attribute metadata, to make them traceable, and to bring historical observations into a form that will be useful to future glaciological studies. As glacier mass balance is an Essential Climate Variable that is highly relevant to global climate observation (Bojinski et al., 2014), it should be a priority to enhance the reliability of the corresponding datasets. Indeed, they form the basis for understanding the link between glaciers and climate.

The rescue and documentation of historical point mass balance data has strengthened the data coverage and documentation that is available for the Swiss Alps. The enhanced and extended basis of (sub-)seasonal point mass balance data for 63 individual Swiss glaciers, covering a period of almost 140 years, represents a globally unequalled dataset. The present work has indicated that tapping into the raw observations of point mass balance is a laborious task. Although about two thirds of the observations



have been previously available in digital form, their quality was often unknown. By re-locating the source of the data in historic, handwritten or printed sources, and by re-evaluating digital sources, metadata for all more than 60'000 entries were added. Data series previously not available digitally were compiled for two dozen additional glaciers (mostly short series) based on scattered information. We consider the new data format, with complete metadata and uncertainties specified for each measurement point as an important basis for the future documentation and storage of mass balance data.

Century-scale direct observations of seasonal point mass balance indicate that the substantial acceleration of glacier mass loss in the Alps since the 1980s is driven by increased summer melting, whereas winter accumulation remained stable. Sites at low elevation show more pronounced temporal shifts in mass balance than sites located in the accumulation zone. More than 800 direct measurements of snow density on glaciers distributed over one century show that no significant temporal change in density occurred, neither for the winter nor the annual period. The average value of end-of-summer snow density is found to be $539\pm86\,\mathrm{kg\,m^{-3}}$, and $429\pm52\,\mathrm{kg\,m^{-3}}$ for end-of-winter snow density. Uncertainties in point mass balance measurements based on a combination of the individual components and the involved assumptions are estimated to be between 0.05 and 0.14 m w.e. for annual mass balance, and between 0.17 and 0.22 m w.e. for winter balance. This corresponds to an average relative uncertainty of 3% for measuring local annual ablation, 13% for annual accumulation, and 16% for the observation of local winter snow accumulation.

Both in previous compilations of Swiss glacier mass balance data, as well as for point mass balances available from glacier monitoring programmes worldwide (WGMS, 2020), only final measurement values are typically available with no or few metadata. This importantly limits the potential scope of the datasets as the quality of the individual data point can vary greatly, depending on field techniques and conditions. Furthermore, problems with individual measurements, as well as approaches to supplement potentially missing information (e.g. estimated snow density) cannot be identified. It is thus crucial to include metadata about the circumstances of data acquisition, the measurement technique and the data quality to be able to provide sound uncertainty estimates. Clearly, the latter is of cardinal importance for the further usage of the dataset. We encourage similar initiatives to be performed in other monitoring programmes to rescue and document long-term point glacier mass balance measurements as the risk to lose access to historic observations, e.g. with a change in responsible scientists or with a shift in data formats, is considerable.

*Data availability.* All data are available under https://doi.glamos.ch/data/massbalance_point/massbalance_point_2021_r2021.html (GLAMOS, 2021).

*Author contributions.* L.G. and C.K. digitized archived data and performed quality checks. L.G. and M.H. performed the data analysis and wrote the paper. A.B. and M.H. provided previous digital datasets. E.H. provided essential support in data management. The study was conceived and supervised by M.H., A.B. and D.F. All authors commented on the manuscript and contributed to the writing.



*Competing interests.* The authors declare no competing interests.

*Acknowledgements.* We would like to acknowledge MeteoSwiss in the framework of the Global Climate Observing System (GCOS) Switzerland for providing project funding. We also thank the innumerous glaciologists acquiring mass balance measurements on Swiss glaciers over the past century, to name just a few with whom we were in direct contact in the frame of this project: A. Linsbauer, O. Langenegger, M. Funk, G. Kappenberger, U. Steinegger, J. Landmann, M. Fischer, H. Machguth, N. Salzmann, M. Zemp.






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
