# Peer review of "Rescue and homogenisation of 140 years of glacier mass balance data in Switzerland"

_Earth System Science Data, 2022_

## Author Response (AR1)

**Revision of "Rescue and homogenisation of 140 years of glacier mass balance data in Switzerland" submitted to ESSD**

Lea Geibel, Matthias Huss, Claudia Kurzböck, Elias Hodel, Andreas Bauder, and Daniel Farinotti

Dear Editor,

We would like to thank the two reviewers for the insightful comments on our manuscript. All points raised were answered and discussed in our response letter below.

All reviewer comments are pasted below. Answers are given in blue, and the relevant revised text in quotation marks.

On behalf of all co-authors,

Matthias Huss

**Comments by Reviewer 1**

Geibel et al (2022) through data rescue, detailed metadata documentation, and record homogenization provide an enhanced and valuable record of Swiss glacier mass balance. This provides both a better record and a good template for such endeavors in other locations. Having worked with such data and produced data that needs this attention, I can attest to the difficulties they systematically overcame to produce a usable and sharable record. The specific comments below are all minor. They aim is to provide further clarity both to the data analysis, big picture trends and potential simplification to the process.

**Answer:**

We would like to acknowledge the reviewer for the constructive review and the helpful comments.
* * *
32: Provide WGMS (2021) reference.

**Answer:**

Done.
* * *
72: "..of an extensive glacier monitoring effort/program/network".

**Answer:**

Thanks, done.
* * *
73: reword "To date mass balance measurements have been performed on more than 60 individual glaciers, most are short time series with 20 glaciers having been monitored for at least three decades."

**Answer:**

Thanks, done.
* * *
127: Is it worth explaining how potential errors can be avoided/identified? "Beginning from a point of known depth such as the snowline or from a snowpit. Measuring at a consistent interval and using the average of 2 or 3 probes within 25 m (Pelto et al, 2013).

**Answer:**

Yes, excellent suggestion. We fully agree with both strategies suggested which have indeed been used in the context of the described series.

Revised text:
«Information from a location with known snow depth, e.g. from measurements in a snow pit, can support identifying the last year's late-summer horizon. Furthermore, 2-3 probings within a radius of 25 m may help detecting erroneous results, and the average of the repeated measurements better accounts for the effect of local surface roughness (see e.g. Pelto et al., 2013).»
* * *
140: Is it worth noting here, or just near line 250, that both end of the winter snowpack and end of summer snowpack density vary from point to point but have a relatively limited mean range.

**Answer:**

Good suggestion, done.

Revised text:
«Snow/firn density varies from point to point but generally within a relatively limited range per glacier and a given date (e.g. Machguth et al. 2006; McGrath et al. 2015; Sold et al., 2016). »
* * *
168: How many of these were along consistent transects where the general route is known?

**Answer:**

This can, unfortunately, not be generally stated. At earlier times, the location of measurement sites was maintained using the angle to terrain markers, e.g. prominent peaks. As the number of individual measurements is relatively limited, one cannot speak of "transects" as such.
* * *
250: I agree with your approach to documenting and reporting density observations as well as examination of their variation in Figure 4. In terms of the broader import of not having a density observation, is this worth more context? You note the variation in density, does this apply to just point or the mean for a glacier? For most glaciers where detailed observations exist the variation in end of winter and end of summer snowpack density is limited. Fausto et al (2018) found that on the GIS snow density within 0.1 m of the surface had an average value of 315 $kgm^{-3}$ + 44 $kg\ m^{-3}$. Further they found insignificant annual air temperature dependency and suggested using a constant density was likely more appropriate for modelling than modelling surface density. On alpine glaciers density measurements for snow during the accumulation season have limited relation to elevation or snow depth (Machguth et al. 2006; McGrath et al. 2015; Sold et al. 2016). By mid-summer on temperate glaciers the density of retained accumulation has a similar behavior approaching a consistent mean value for specific glaciers and icefields that are between 550 and 600 $kgm^{-3}$ across western North America (Bidlake et al. 2010; Pelto et al. 2013; Beedle et al. 2014; Pelto, et al. 2019).

**Answer:**

Thanks for this detailed comment. All our assessments refer to individual points, and not to the glacier-scale for which extrapolation/modelling would need to be involved, i.e. moving away from the pure data analysis presented in this paper. We agree that the spatial variations in density (both winter and annual period) are likely to be limited for given glaciers and survey dates. This is also indicated by the minor dependence of the density on terrain elevation (see Fig. 4). However, the data set is not extensive enough to provide more detailed insights on spatial density variations at the glacier scale in a more general way.

In response to this comment, we now provide some more introduction and context into the spatio-temporal variations in density along the lines suggested by the reviewer.

Revised text:

«Previous studies in different climatic settings have indicated that, for a given date, spatial variations in snow density are relatively limited at the scale of individual glaciers (Machguth et al., 2006; Beedle et al., 2014; Sold et al., 2016; Fausto et al., 2018; Pelto et al., 2019). This is true for both the end of winter and late summer. »
* * *
326: How many measurements were complemented or corrected for missing information?

**Answer:**

The absolute number of such individual updates/corrections to existing data sets is very limited (in the order of a few 100) but metadata were added for all entries. The sentence was somewhat misleading and has now been reformulated.

Revised text:
«they were revisited, missing information was complemented where necessary, and metadata for each entry was added.»
* * *
330: Do you have a specific example where the intermediate measurements are valuable, such as many on a specific glacier or during a specific time interval?

**Answer:**

Such an example is now provided.

Revised text:
«For example, several summer seasons of continuous daily ablation data are available for Rhonegletscher providing information on short-term glacier mass changes (Landmann et al., 2021).»
* * *
370: Figure 6 is incredibly valuable. I would encourage using a mechanism to expand the lateral and vertical extent for glaciers with more than 10 years of record. Is a landscape mode for a page allowed/usable in ESSD to accommodate a particularly wide figure?

**Answer:**

Indeed, we did a lot of experimenting with this figure to optimally use the available space. Landscape was also tested but is not an option as the figure would then need to stretch over several pages to actually provide an enlarged visibility. We therefore need to refer the reader to the possibility to zoom in, or to directly download our data set.
* * *
408: Indicate the timing of this transition from accumulation to ablation at these two sites.

**Answer:**

Done.

Revised text:
«Some long-term measurement sites have also transited from the accumulation to the ablation area between 1980 and 2000 (e.g. Clariden Lower site, Silvretta "BU") ... »
* * *
410: For visualization of the trends in winter and summer I suggest adding a figure or panel with all of the summer record and winter records of the glaciers in Figure 7 on the same plot. This allows seeing how similar trends are.

**Answer:**

We agree that such an additional figure might be valuable. However, a direct comparison (i.e. in the same panel) of point mass balance series at different elevations is difficult both in terms of clarity/visibility, as well as conceptually. This is why we have performed a dedicated analysis to analyze differences between the sites and to compare the trends. This is visualized in Figure 8. We therefore rather would like to refer the reader to that analysis and the corresponding figure (Fig. 8) for more insights into this aspect.
* * *
429: That summer balance changes are the key has been noted in many alpine regions around the world, is that worth noting here? WGMS GGCB #4 (2021) illustrates seasonal balance for the regions with long balance records from more than a couple glaciers 3.1 (Alaska), 3.2 (Western North America), 3.7 (Scandinavia) and 3.8 (Central Europe), all show this declining summer balance trend and limited winter balance trend.

**Answer:**

Good suggestion. A corresponding statement has been added.

Revised text:
«The dominant role of the recent increase in ablation rates in driving accelerated glacier mass loss has also been documented on glaciers worldwide (WGMS, 2021).»
* * *
452: Not sure why this would be expected in a snowpack that is at 0 C. There is certainly a documented evolution of density to a through mid-summer, but after that it is more about removable of thickness than any densification/refreezing processes. No need to address unless you see value in addressing based on local observations.

**Answer:**

This is a very valid objection against our formulation, and we agree that such a density increase at the annual scale cannot actually be expected under climate change. The statement has been reformulated correspondingly.

Revised text:
«The absence of trends indicates that climate change has not (yet) resulted in a detectable impact on the snow density on glaciers in the Alps - neither for winter, nor the annual period - although the former might be expected from higher winter temperatures. »
* * *
492: The lack of temporal change in density and limited EOS and EOW density all argue that applying a standard density would be appropriate to substitute for in-situ observations. Could reference Sold et al. (2016) here since they had a similar result with no trends in density spatially or with altitude.

**Answer:**

While our results clearly indicate that for the annual scale a standard density is appropriate (and has been used in the present study), Fig. 4A and Fig. 4C show that for end-of-winter snow density the dependency on the timing (day of year) of the measurement and snow depth is significant. As this is the Conclusion Section, we would like to stay general and rather would avoid providing additional references. Nevertheless we have added a statement to emphasize this aspect.

Revised text:
«No variables explaining variations in end-of-summer snow density were detected, while end-of-winter snow density was found to depend on the timing of the observation and snow depth.»
* * *
References:

…

**Answer:**

Thanks for the extensive suggestions of appropriate reference. All of them have now been included in the revised version of the manuscript.

**Comments by Reviewer 2**

The manuscript describes the compilation of an extensive glacier point mass balance dataset for 63 Swiss glaciers, with data stemming from the years 1884 - 2020 CE. Multiple archives were sourced for original data, and metadata (including attributes for data quality and uncertainties) was added for all entries in the data compilation. The dataset is of highest value not only in the context of Swiss glacier mass balance studies. Echoing the authors and emphasizing their suggestion, the structure of the dataset described here should indeed be used a template for similar compilation, homogenization and rescue of glacier mass balance data from other regions around the world.

**Answer:**

We would like to acknowledge the reviewer for the positive review and the helpful comments.
* * *
L 89-90: The sentence "The annual observations…" is obsolete.

**Answer:**

Removed.
* * *
L 95: Could a reference be provided as an example where short term observations have been used for mass balance model calibration and validation?

**Answer:**

References are now provided: Braithwaite, 2009; Litt et al., 2019; Landmann et al., 2021.
* * *
L 253: check wording /grammar ("with for a…"), and, do you mean w.e. > 0.1 m (instead of "beyond")?

**Answer:**

Thanks for this remark. The sentence has been edited correspondingly.
* * *
L 273-274: In line 274 the average of all end-of summer snow density is given as 539 km/m3. Is this the same as the annual density (in contrast to the winter density)? Annual snow density has a slightly different value in Table 2, please check. Why do you chose to replace a missing density at annual scale (year X prior to 2020) with the average annual snow density over all years, and not the average until that specific year (X)?

**Answer:**

Yes, this value corresponds to the measured density at the annual scale (we never use the term "annual density").

Regarding the slightly different value given here in contrast to Table 2 the explanation is simple: Table 2 shows the period 1951 to 2020 while here (and on line 435 and 492) we refer to the complete data set (1880-2021). Although this only adds relatively few data points, the average is slightly different. To clarify this, we extended the caption of Table 2, emphasizing that only part of the entire data set is shown here.

Furthermore, winter densities shown in Table 2 have been homogenized to correspond to 30 April based on the temporal densification rate to allow direct comparability, while the overall averages stated here and on L 435 and L 492 refer to the raw measurements.

We understand the reviewer's comment regarding the long-term temporal variations in density. This analysis is only presented later in the manuscript. We thus added a reference to these findings (no temporal changes in density) that support our decision to use a constant value. The analysis shown in Fig. 4, as well as in Table 2, clearly indicated that for end-of-summer snow density (or density at the annual scale) neither a dependence on potential explaining variables (date of measurements, elevation, snow depth), nor a temporal trend over the last century can be detected. It would therefore not be appropriate to vary the value supplemented for missing densities as the available data set simply does not provide sufficient information on such potential variations. We therefore decided to choose the most robust solution, i.e. to use the average of all measurements.

Revised text:
«In addition, no long-term temporal changes in density at the annual scale are evident from the data (see Section 4.3 for details). We therefore use the average of all observations of the end-of-summer snow density (539 kg m-3) to provide a density estimate for missing entries at the annual scale.»
* * *
Fig 4: Please explain better what the red lines indicate.

**Answer:**

The caption was extended to provide more information.

Revised text:
«The straight black line corresponds to a linear fit, and bold red lines show average densities for five equally-spaced classes of each analyzed variable for illustration. The latter indicates trends beyond the variability of the individual data points.»
* * *
L 335: Glaciers are not always consistently named, cf. e.g. L 385 (Jöri, Jörigletscher). Also in other places, please check.

**Answer:**

Thanks for making us aware of this. We have gone through the paper again to check for consistency. We decided to not use any short names (i.e. without "glacier" or equivalent) in the text now.
* * *
L 435: Annual snow density, and winter snow density, as given here differ from the values in Table 2, please check.

**Answer:**

See explanation above.

Revised text:
«Dedacal averages of observed snow densities both at the annual scale rho_ann and for the winter period rho_win between 1951 and 2020.»
* * *
L 492: cf. L 435

**Answer:**

See above.

---

## Author Response (AR2)

**Revision of "Rescue and homogenisation of 140 years of glacier mass balance data in Switzerland" submitted to ESSD**

Lea Geibel, Matthias Huss, Claudia Kurzböck, Elias Hodel, Andreas Bauder, and Daniel Farinotti

Dear Editor,

Thanks for the comment regarding Fig. 5. This is a good suggestion and has fully been implemented, including a minor adaptation of the Caption. The revised pdf has now been uploaded.

On behalf of all co-authors,

Matthias Huss